# Learning and Planning with a Semantic Model

## Abstract

Building deep reinforcement learning agents that can generalize and adapt to unseen environments remains a fundamental challenge for AI. This paper describes progresses on this challenge in the context of man-made environments, which are visually diverse but contain intrinsic semantic regularities. We propose a hybrid model-based and model-free approach, *LEArning and Planning with Semantics (LEAPS)*, consisting of a multi-target sub-policy that acts on visual inputs, and a Bayesian model over semantic structures. When placed in an unseen environment, the agent plans with the semantic model to make high-level decisions, proposes the next sub-target for the sub-policy to execute, and updates the semantic model based on new observations. We perform experiments in visual navigation tasks using House3D, a 3D environment that contains diverse human-designed indoor scenes with real-world objects. LEAPS outperforms strong baselines that do not explicitly plan using the semantic content.

## 1 Introduction

Deep reinforcement learning (DRL) has undoubtedly witnessed strong achievements in recent years (Silver et al., 2016; Mnih et al., 2015; Levine et al., 2016). However, training an agent to solve tasks in a new unseen scenario, usually referred to as its *generalization ability*, remains a challenging problem (Geffner, 2018; Lake et al., 2017). In model-free RL, the agent is trained to reactively make decisions from the observations, e.g., first-person view, via a black-box policy approximator. However the generalization ability of agents trained by model-free RL is limited, and is even more evident on tasks that require extensive planning (Tamar et al., 2016; Kansky et al., 2017). On the other hand, model-based RL learns a dynamics model, predicting the next observation when taking an action. With the model, sequential decisions can be made via planning. However, learning a model for complex tasks and with high dimensional observations, such as images, is challenging. Current approaches for learning action-conditional models from video are only accurate for very short horizons (Finn & Levine, 2017; Ebert et al., 2017; Oh et al., 2015). Moreover, it is not clear how to efficiently adapt such models to changes in the domain.

In this work, we aim to improve the generalization of RL agents in domains that involve high-dimensional observations. Our insight is that in many realistic settings, building a pixel-accurate model of the dynamics is not necessary for planning high-level decisions. There are semantic structures and properties that are shared in real-world man-made environments. For example, rooms in indoor scenes are often arranged by their mutual functionality (e.g. , bathroom next to bedroom, dining room next to kitchen). Similarly, objects in rooms are placed at locations of practical significance (e.g. , nightstand next to bed, chair next to table). Humans often make use of such structural priors when exploring a new scene, or when making a high-level plan of actions in the domain. However, pixel-level details are still necessary for carrying out the high-level plan. For example, we need high-fidelity observations to locate and interact with objects, open doors, etc.

Based on this observation, we propose a hybrid framework, *LEArning and Planning with Semantics (LEAPS)*, which consists of a model-based component that works on the semantic level to pursue a high-level target, and a model-free component that executes the target by acting on pixel-level inputs. Concretely, we (1) train model-free multi-target subpolicies in the form of neural networks that take the first-person views as input and sequentially execute sub-targets towards the final goal; (2) build a *semantic* model in the form of a latent variable model that only takes *semantic signals*, i.e., low-dimensional binary vectors, as input and is dynamically updated to plan the next sub-target. LEAPS has following advantages: (1) via model-based planning, generalization ability is improved; (2) by learning the *prior* distribution of the latent variable model, we capture the semantic consistency among the environments; (3) the semantic model can be efficiently updated by posterior inference

when the agent is exploring the unseen environment, which is effective even with very few exploration experiences thanks to the Bayes rule; and (4) the semantic model is lightweight and fully interpretable.

Our approach requires observations that are composed of both pixel-level data and a list of semantic properties of the scene. In general, automatically extracting high-level semantic structure from data is difficult. As a first step, in this work we focus on domains where obtaining semantics is easy. In particular, we consider environments which resemble the real-world and have strong object detectors available (He et al., 2017). An example of such environments is House3D which contains 45k human-designed 3D scenes (Wu et al., 2018). House3D provides a diverse set of scene layouts, object types, sizes and connectivity, which all conform to a consistent "natural" semantics. Within these complex scenes, we tackle navigation tasks within novel indoor scenes. Note that this problem is extremely challenging as the agent needs to reach far-away targets which can only be completed effectively if it can successfully reason about the overall structure of the new scenario. Lastly, we emphasize that although we consider navigation as a concrete example in this work, our approach is general and can be applied to other tasks for which semantic structures and signals are available

Our extensive experiments show that our LEAPS framework outperforms strong model-free RL approaches, even when the semantic signals are given as input to the policy. Furthermore, the relative improvements of LEAPS over baselines become more significant when the targets are further away from the agent's birthplace, indicating the effectiveness of planning on the learned semantic model.

## 2 RELATED WORK

Most deep RL agents are tested in the same training environments (Mirowski et al., 2016), disregarding generalization. While limited, robust training approaches have been proposed to enforce an agent's generalization ability, such as domain randomization (Tobin et al., 2017) and data augmentation by generating random mazes for training (Oh et al., 2017; Parisotto & Salakhutdinov, 2017). In our work, we use a test set of novel unseen environments, where an agent cannot resort to memorization or simple pattern matching to solve the task.

Meta-learning has shown promising results for fast adaptation to novel environments. Methods include learning a good initialization for gradient descent (Finn et al., 2017) or learning a neural network that can adapt its policy during exploration (Duan et al., 2016; Mishra et al., 2017). We propose to learn a Bayesian model over the semantic level and infer the posterior structure via the Bayes rule. Our approach (1) can work even without any exploration steps in a new environment and (2) is interpretable and can be potentially combined with any graph-based planning algorithm.

Our work can be viewed as a special case of hierarchical reinforcement learning (HRL). Unlike other approaches (Vezhnevets et al., 2017; Bacon et al., 2017), in our work high-level planning is performed based on the semantic signals. With orders of magnitudes fewer parameters, our approach is easier to learn compared to recurrent controllers.

LEAPS assumes a discrete semantic signal in addition to the continuous state. A similar assumption is also adopted in (Zhang et al., 2018), where the discrete signals are called "attributes" and used for planning to solve compositional tasks within the same fully observable environment. (Riedmiller et al., 2018) use additional discrete signals to tackle the sparse reward problem. The schema network (Kansky et al., 2017) further assumes that even the continuous visual signal can be completely represented in a binary form and therefore directly runs logical reasoning on the binary states.

For evaluating our approach, we focus on the problem of visual navigation, which has been studied extensively (Leonard & Durrant-Whyte, 1992). Classical approaches build a 3D map of the scene using SLAM, which is subsequently used for planning (Fox et al., 2005). More recently, end-to-end approaches have been applied to tackle various domains, such as maze (Mirowski et al., 2016), indoor scenes (Zhu et al., 2017) and Google street view (Mirowski et al., 2018). Evidently, navigation performance deteriorates as the agent's distance from the target increases (Zhu et al., 2017; Wu et al., 2018). To aid navigation and boost performance, auxiliary tasks (Mirowski et al., 2016; Jaderberg et al., 2016) are often introduced during training. Another direction for visual navigation is to use a recurrent neural network and represent the memory in the form of a 2D spatial map (Khan et al., 2018; Parisotto & Salakhutdinov, 2017; Tamar et al., 2016; Gupta et al., 2017) such that a differentiable planning computation can be performed on the spatial memory. Our approach considers more general graph structures beyond dense 2D grids and captures relationships between *semantic* signals, which we utilize as an informative latent structure in semantically rich environments like House3D.

Similar to our work, Savinov *et al.* (Savinov et al., 2018) constructs a graph of nodes corresponding to different locations of the environment. However, they rely on a pre-exploration step within the test scene and build the graph completely from the pixel space. In LEAPS, we use semantic knowledge and learn a prior over the semantic structures that are shared across real-world scenes. This allows us to directly start solving for the task at hand without any exploratory steps.

# 3 BACKGROUND

We assume familiarity with standard DRL notations. Complete definitions are in Appendix A.

**Environment:** We consider a *contextual Markov decision process* (Hallak et al., 2015) $E(c)$ defined by $E(c) = (\mathcal{S}, \mathcal{A}, P(s'|s, a; c), r(s, a; c))$. Here $c$ represents the objects, layouts and any other *semantic* information describing the environment, and is sampled from $\mathcal{C}$, the distribution of possible semantic scenarios. For example, $c$ can be intuitively understood as encoding the complete map for navigation, or the complete object and obstacle layouts in robotics manipulations, not known to the agent in advance, and we refer to them as the *context*.

**Semantic Signal:** At each time step, the agent observes from $s$ a tuple $(s_o, s_s)$, which consists of: (1) a high-dimensional observation $s_o$, e.g., the first person view image, and (2) a low-dimensional discrete *semantic signal* $s_s$, which encodes semantic information. Such signals are common in AI, e.g., in robotic manipulation tasks $s_s$ indicates whether the robot is holding an object; for games it is the game status of a player; in visual navigation it indicates whether the agent reached a landmark; while in the AI planning literature, $s_s$ is typically a list of *predicates* that describe binary properties of objects. We assume $s_s$ is provided by an oracle function, which can either be directly provided by the environment or extracted by some semantic extractor.

**Generalization:** Let $\mu(a|\{s^{(t)}\}_t; \theta)$ denote the agent's policy parametrized by $\theta$ conditioned on the previous states $\{s^{(t)}\}_t$. The objective of *generalization* is to train a policy on training environments $\mathcal{E}_{\text{train}}$ such that the accumulative reward $R(\mu(\theta); c)$ on test set $\mathcal{E}_{\text{test}}$ is maximized.

# 4 LEARNING AND PLANNING WITH A SEMANTIC MODEL

The key motivation of LEAPS is the fact that while each environment can be different in visual appearances, there are structural similarities between environments that can be captured as a probabilistic graphical model over the semantic information. On a high level, we aim to learn a Bayesian model $\mathbf{M}^\star(\mathcal{D}, c)$ that captures the semantic properties of the context $c$, from the agent's exploration experiences $\mathcal{D}$. Given a new environment $E(c')$, the agent computes the posterior $P(c'|\mathcal{D}', \mathbf{M}^\star)$ for the unknown context $c'$ via the learned model $\mathbf{M}^\star$ and its current experiences $\mathcal{D}'$. This allows the agent to plan according to its belief of $c'$ to reach the goal more effectively. Thanks to the Bayes rule, this formulation allows probabilistic inference even with *limited* (or even no) exploration experiences.

Learning an accurate and complete Bayesian model $\mathbf{M}^\star(\mathcal{D}, c)$ can be challenging. We learn an *approximate* latent variable model $\mathbf{M}(y, z; \psi)$ parameterized by $\psi$ with observation variable $y$ and latent variable $z$ that *only* depend on the *semantic signal* $s_s$. Suppose we have $K$ different semantic signals $T_1, \ldots, T_K$ and $s_s \in \{0, 1\}^K$ where $s_s(T_k)$ denotes whether the $k$th signal $T_k$ (e.g., landmarks in navigation) is reached or not. Assuming $T_1$ is the final goal of the task, from any state $s$, we want to reach some final state $s'$ with $s'_s(T_1) = 1$. In this work, we consider navigation as a concrete example, which can be represented as reaching a state where a desired semantic signal becomes 'true'. We exploit the fact that navigation to a target can be decomposed into reaching several way points on way to the target, and therefore can be guided by planning on the semantic signals, i.e., arrival at particular way points.

## 4.1 THE SEMANTIC MODEL

Note that there can be $2^K$ different values for $s_s$. For efficient computation, we assume *independence* between different semantic signals $T_k$: we use a binary variable $z_{i,j}$ to denote whether some state $s'$ with $s'_s(T_j) = 1$ can be "*directly reached*", i.e., by a few exploration steps, from some state $s$ with $s_s(T_i) = 1$, *regardless* of other signals $T_k \notin \{T_i, T_j\}$. In addition, we also assume *reversibility*, i.e., $z_{i,j} = z_{j,i}$, so only $K(K-1)/2$ latent variables are needed. Before entering the unknown environment, the agent does not know the true value of $z_{i,j}$, but holds some prior belief $P(z_{i,j})$, defined by $z_{i,j} \sim \text{Bernoulli}(\psi_{i,j}^{\text{prior}})$, where $\psi_{i,j}^{\text{prior}}$ is some parameter to be learned. After some exploration steps, the agent receives a noisy observation $y_{i,j}$ of $z_{i,j}$, i.e., whether a state $s'$ with

$s'_s(T_j) = 1$ is reached. We define the observation model $P(y_{i,j}|z_{i,j})$ as follows:

$$y_{i,j} \sim \begin{cases} \text{Bernoulli}(\psi^{\text{obs}}_{i,j,0}) & \text{if } z_{i,j} = 0 \\ \text{Bernoulli}(1 - \psi^{\text{obs}}_{i,j,1}) & \text{if } z_{i,j} = 1 \end{cases} \tag{1}$$

At any time step, the agent hold an overall belief $P(z|\mathcal{Y})$ of the semantic structure of the unknown environment, based on its experiences $\mathcal{Y}$, namely the samples of $y$.

## 4.2 Combining the Semantic Model with Multi-Target Sub-policies

**Multi-target sub-policies:** With our semantic model, we correspondingly learn multi-target sub-policies $\mu(a|\{s_o^{(t)}\}_t; T_i, \theta)$ taking $s_o$ as input such that $\mu(T_i, \theta)$ is particularly trained for sub-target $T_i$, i.e., reaching a state $s'$ with $s'_s(T_i) = 1$. Hence the semantic model can be treated as a model-based *planning module* that picks an intermediate sub-target for the sub-policies to execute so that the final target $T_1$ can be reached with the highest probability.

**Inference and planning on M:** We assume the agent explores the current environment for a *short* horizon of $N$ steps and receives semantic signals $s_s^{(1)}, \ldots, s_s^{(N)}$. Then we compute the bit-OR operation over these binary vectors $B = s_s^{(1)}$ `OR` $\ldots$ `OR` $s_s^{(N)}$. By the reversibility assumption, for $T_i$ and $T_j$ with $B(T_i) = B(T_j) = 1$, we know that $T_i$ and $T_j$ are "directly reachable" for each other, namely a sample of $y_{i,j} = 1$, and otherwise $y_{i,j} = 0$. Combining all the history samples of $y$ and the current batch from $B$ as $\mathcal{Y}$, we can perform posterior inference $P(z|\mathcal{Y})$ by the Bayes rule. By the independence assumption, we can individually compute the belief of each latent variable $z_{i,j}$, denoted by $\hat{z}_{i,j} = P(z_{i,j}|\mathcal{Y}_{i,j})$. Given the current beliefs $\hat{z}_{i,j}$, the current semantic signals $s_s$ and the goal $T_1$, we search for an optimal plan $\tau^* = \{\tau_0, \tau_1, \ldots, \tau_{m-1}, \tau_m\}$, where $\tau_i \in \{1 \ldots K\}$ denotes an index of concepts and particularly $\tau_m = 1$, so that the joint belief along the path from some current signal to the goal is maximized:

$$\tau^\star = \arg\max_\tau s_s(T_{\tau_0}) \prod_{t=1}^m \hat{z}_{\tau_{t-1}, \tau_t}. \tag{2}$$

After obtaining $\tau^\star$, we execute the sub-policy for the next sub-target $T_{\tau_1^\star}$, and then repeatedly update the model and replan every $N$ steps.

## 4.3 Learning the Semantic Model

The model parameters $\psi$ have two parts: $\psi^{\text{prior}}$ for the prior of $z$ and $\psi^{\text{obs}}$ for the noisy observation $y$. Note that $\psi^{\text{obs}}$ is related to the performance of the sub-policies $\mu(\theta)$: if $\mu(\theta)$ has a high success rate for reaching sub-targets, $\psi^{\text{obs}}$ should be low; when $\mu(\theta)$ is poor, $\psi^{\text{obs}}$ should be higher (cf. Eq. (1)).

**Learning $\psi^{\text{prior}}$:** We learn $\psi^{\text{prior}}$ from $\mathcal{E}_{\text{train}}$. During training, for each pair of semantic signals $T_i$ and $T_j$, we run *random explorations* from some state $s$ with $s(T_i) = 1$. If eventually we reach some state $s'$ with $s'(T_j) = 1$, we consider $T_i$ and $T_j$ are reachable and therefore a *positive* sample $z_{i,j} = 1$; otherwise a *negative* sample $z_{i,j} = 0$. Suppose $\mathcal{Z}$ denotes the samples we obtained for $z$ from $\mathcal{E}_{\text{train}}$. We run maximum likelihood estimate for $\psi^{\text{prior}}$ by maximizing $L_{\text{MLE}}(\psi^{\text{prior}}) = P(\mathcal{Z}|\psi^{\text{prior}})$.

**Learning $\psi^{\text{obs}}$:** There is no direct supervision for $\psi^{\text{obs}}$. However, we can evaluate a particular value of $\psi^{\text{obs}}$ by policy evaluation on the validation environments $\mathcal{E}_{\text{valid}}$. We optimize the accumulative reward $L_{\text{valid}}(\psi^{\text{obs}}) = \mathbb{E}_{E(c) \in \mathcal{E}_{\text{valid}}}[R(\mu(\theta), M(\psi); c)]$, with the semantic model $M(\psi)$. Analytically optimizing $L_{\text{valid}}$ is hard. Instead, we apply local search in practice to find the optimal $\psi^{\text{obs}}$.

## 4.4 Learning the LEAPS agent

The LEAPS agent consists of two parts, the multi-target sub-policy $\mu(T_i, \theta)$ and the semantic model $M(\psi)$. Learning the multi-target sub-policies can be accomplished by any standard deep RL method on $\mathcal{E}_{\text{train}}$. For the semantic model, learning $\psi^{\text{prior}}$ does not depend on the sub-policies and can be reused even with different sub-policies; $\psi^{\text{obs}}$ depends on the sub-policies so it should be learned after $\mu(T_i, \theta)$ is obtained.

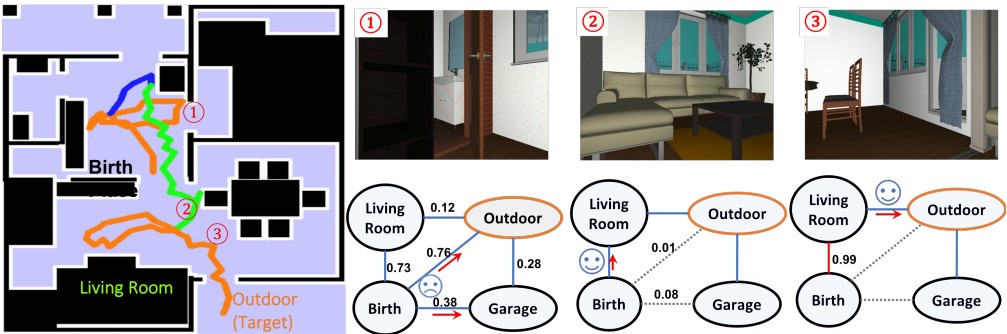

Figure 1: Visualization of learned semantic prior of $\mathbf{M}(\psi)$: the most and least likely nearby rooms for dining room (L), bedroom (M) and outdoor (R), with numbers denoting $\psi^z$, i.e., the probability of two rooms connecting to each other.

Figure 2: Example of a successful trajectory. The agent is spawned inside the house, targeting "outdoor". **Left**: the 2D top-down map with sub-target trajectories ("outdoor" – orange; "garage" – blue; "living room" – green); **Right, 1st row**: RGB visual image; **Right, 2nd row**: the posterior of the semantic graph and the proposed sub-targets (red arrow). Initially, the agent starts by executing the sub-policy "outdoor" and then "garage" according to the prior knowledge (**1st graph**), but both fail (top orange and blue trajectories in the map). After updating its belief that garage and outdoor are not nearby (grey edges in the **2nd graph**), it then executes the "living room" sub-policy with success (red arrow in the **2nd graph**, green trajectory). Finally, it executes "outdoor" sub-policy again, explores the living room and reaches the goal (**3rd graph**, bottom orange trajectory).

## 5 ROOMNAV: A 3D NAVIGATION TASK FOR RL GENERALIZATION

RoomNav is a concept driven navigation task based on the House3D environment (Wu et al., 2018). In RoomNav, the agent is given a concept target, i.e., a room type, and needs to navigate to find the target room. RoomNav pre-selected a fixed set of target room types and provides a training set of 200 houses, a testing set of 50 houses and a small validation set of 20 houses.

**Semantic signals:** We choose the $K = 8$ most common room types as our semantic signals, such that $s_s(T_i)$ denotes whether the agent is currently in a room with type $T_i$[1]. When given a target $T_i$, reaching a state $s$ with $s_s(T_i) = 1$ becomes our final goal. House3D provides bounding boxes for rooms, which can be directly used as the oracle for semantic signals. But in practice, we only use these oracle signals to train a room type detector and use this detector to extract semantic information during evaluation. Details can be found in the beginning part of Sec. 6.

**The semantic model and sub-policies:** In navigation, the reachability variable $z_{i,j}$ can naturally represent the connectivity between room type $T_i$ and room type $T_j$[2]. We run random explorations in training houses between rooms to collect samples for learning $\psi^{\text{prior}}$. For learning $\psi^{\text{obs}}$, we perform a grid search and evaluate on the validation set. For sub-policies, we learn target driven LSTM policies by A3C (Mnih et al., 2016) with shaped reward on $\mathcal{E}_{\text{train}}$. More details are in Appendix. G.

## 6 EXPERIMENTS

In this section, we experiment on RoomNav and try to answer the following questions: **(1)** Does the learned prior distribution capture meaningful semantic consistencies? **(2)** Does our LEAPS agent generalize better than the model-freel RL agent that *only* takes image input? **(3)** Our LEAPS

---

[1]We also treat $s_s = \mathbf{0}$ as a special semantic signal. So $\mathbf{M}$ actually contains $K + 1$ signals.
[2]A house can have multiple rooms of the same type. But even this simplification improves generalization.

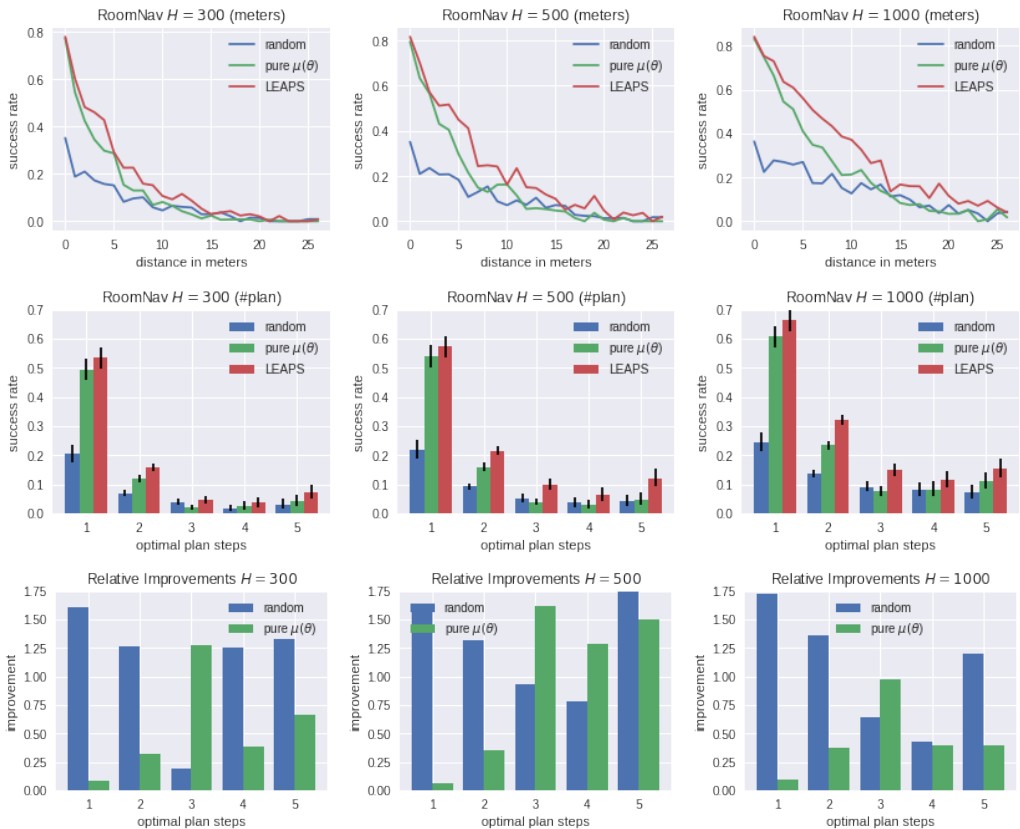

Figure 3: Comparison in success rate with model-free baselines (Sec. 6.2). We evaluate performance of random policy (blue), model-free RL baseline (pure $\mu(\theta)$, green) and our LEAPS agent (red), with increasing horizon $H$ from left to right (**left**: $H = 300$; **middle**: $H = 500$; **right**: $H = 1000$). Each row shows a particular metric. **Top row**: success rate (y-axis) w.r.t. the distance in meters from the birthplace to target room (x-axis); **middle row**: success rate with confidence interval (y-axis) w.r.t. the shortest planning distance in the ground truth semantic model (x-axis); **bottom row**: relative improvement of LEAPS over the baseline (y-axis) w.r.t. the optimal plan distance (x-axis). As the number of planning computations, i.e., $H/N$, increases (from left to right), LEAPS agent outperforms baselines more. LEAPS also has higher relative improvements, i.e., *40% to 180%*, for targets requiring more semantic planning computations (i.e., plan-steps > 2).

agent takes additional semantic signals as input. How does LEAPS compare to other model-free RL approaches that *also* take the semantic signals as part of the inputs but in a different way from our semantic model? For example, what about replacing our semantic model with a complicated RNN controller? **(4)** What do our LEAPS agents perform under some other metric considering the episode length? We consider a recently proposed metric, *SPL* (Anderson et al., 2018) in Sec. 6.4.

**Semantic Signals:** All the semantic signals fed to our semantic model at *test time* are extracted by a CNN room type detector, which is trained on the (noisy) oracle semantic information provided by the House3D on $\mathcal{E}_{\text{train}}$ and validated on $\mathcal{E}_{\text{valid}}$. During training, all the approaches directly use the oracle semantic signals. More details are in Appendix J and Appendix D.

## 6.1 The Learned Prior of the Semantic Model

We visualize the learned prior $P(z|\psi^{\text{prior}})$ in Fig. 1 with 3 room types and their most and least likely connected rooms. The learned prior indeed captures reasonable relationships: bathroom is likely to connect to a bedroom; kitchen is often near a dining room while garage is typically outdoor.

## 6.2 COMPARISON WITH MODEL-FREE RL BASELINES

We follow measure the testing success rate of different agents under various horizon lengths on $\mathcal{E}_{\text{test}}$. More details are in Appendix C and F. We compare our LEAPS agent with two baselines (1) random policy (denoted by "random") and (2) model-free RL agent that only takes in image input $s_o$ and executes $\mu(T_i, \theta)$ throughout the episode (denoted by "pure $\mu(\theta)$"). For LEAPS agent, we set $N = 30$, i.e., update the semantic model every 30 steps. We experiment on horizons $H = 300, 500, 1000$ and evaluate the success rate and relative improvements of our LEAPS agent over the baselines in Fig. 3. As the number of planning computations, $H/N$, increases, our LEAPS agent outperforms the baselines more significantly in success rate. Note that since targets of plan-steps 1 do not require any planning computations, hence it is as expected that LEAPS does not improve much over the pure policy. The best relative improvements are achieved for targets neither too faraway nor too close, i.e., plan steps equal to 3 or 4. Interestingly, we observe that there is a small success rate increase for targets that are 5 plan steps away. We suspect that this is because it is rare to see houses that has a diameter of 5 in the semantic model (imagine a house where you need to go through 5 rooms to reach a place). Such houses may have structural properties that makes navigation easier. Fig. 2 shows an example of a success trajectory of our LEAPS agent. We visualize the progression of the episode, describe the plans and show the updated graph after exploration.

## 6.3 COMPARING TO SEMANTIC-AWARE AGENTS WITHOUT A GRAPH REPRESENTATION

Here we consider two semantic-aware agents that also takes the semantic signals as input.

**Semantic augmented agents:** We train new sub-policies $\mu_s(\theta_s)$ taking both $s_o$ and $s_s$ as input.

**HRL agents with a RNN controller:** Note that updating and planning on $M$ (Eq. 2) only depend on (1) the current semantic signal $s_s$, (2) the target $T_i$, and (3) the accumulative bit-OR feature $B$. Hence, we fixed the same set of sub-policies $\mu(\theta)$ used by our LEAPS agent, and train an LSTM controller with 50 hidden units on $\mathcal{E}_{\text{train}}$ that takes all the necessary semantic information, and produce a sub-target every $N$ steps. Training details are in Appendix I. Note that the only difference between our LEAPS agent and this HRL agent is the *representation* of the planning module. The LSTM controller has access to exactly the same semantic information as our model $M$ and uses a much more complicated neural model. Thus we expect it to perform competitively to our LEAPS agent.

The results are shown in Fig. 4, where our LEAPS agent outperforms both baselines. The semantic augmented policy $\mu_s(\theta_s)$ does not improve much on the original $\mu(\theta)$. For the HRL agent with an LSTM controller, the LEAPS agent achieves higher relative improvements for targets requiring more planning computations (i.e., plan-steps > 1), and also has the following advantages: (1) $M$ can be learned more efficiently with much fewer parameters: an LSTM with 50 hidden units has over $10^4$ parameters while $M(\psi)$ only has 38 parameters[3]; (2) $M$ can adapt to new sub-policies $\mu(\theta')$ with little fine-tuning ($\psi^{\text{prior}}$ remains unchanged) while the LSTM controller needs to re-train; (3) the model $M$ and the planning procedure are fully interpretable.

## 6.4 EVALUATION UNDER METRICS CONSIDERING EPISODE LENGTH

In the previous evaluation, we only consider the metric *success rate* under different horizons. Indeed, another informative metric will be the episode length. There are two important factors: (1) we would expect a better navigation agent to finish the semantic task in a shorter amount of actions; (2) we should assign more credits to the agent when it finishes a hard episode while less credits when finishing an easy one. Recently, there is a new evaluation metric proposed for embodied navigation agent by Anderson et al. (2018) capturing both these two factors, i.e., the *Success weighted by Path Length (SPL)* metric. SPL is a function considering both success rate and the path length to reach the goal from the starting point defined by $\frac{1}{N} \sum_i S_i \frac{L_i}{\max(L_i, P_i)}$, where $N$ is total episodes evaluated, $S_i$ indicates whether the episode is success or not, $L_i$ is the ground truth shortest path distance in the episode, $P_i$ is the number of steps the agent actually took.

We evaluate the performance of LEAPS agents against all baseline agents in the metric of both success rate and SPL in Figure 5. Our LEAPS agent has the highest average SPL (rightmost column) with a big margin over all baseline agents in all the cases. Notably, the margin in SPL is much more significant than the margin in pure success rate. More importantly, as the horizon increases, namely, more planning computations allowed, the SPL margin of LEAPS over the best remaining

---

[3]We assign the same value to all $\psi^{\text{obs}}_{i,j,c}$ for each $c \in \{0, 1\}$. See more in Appendix H.

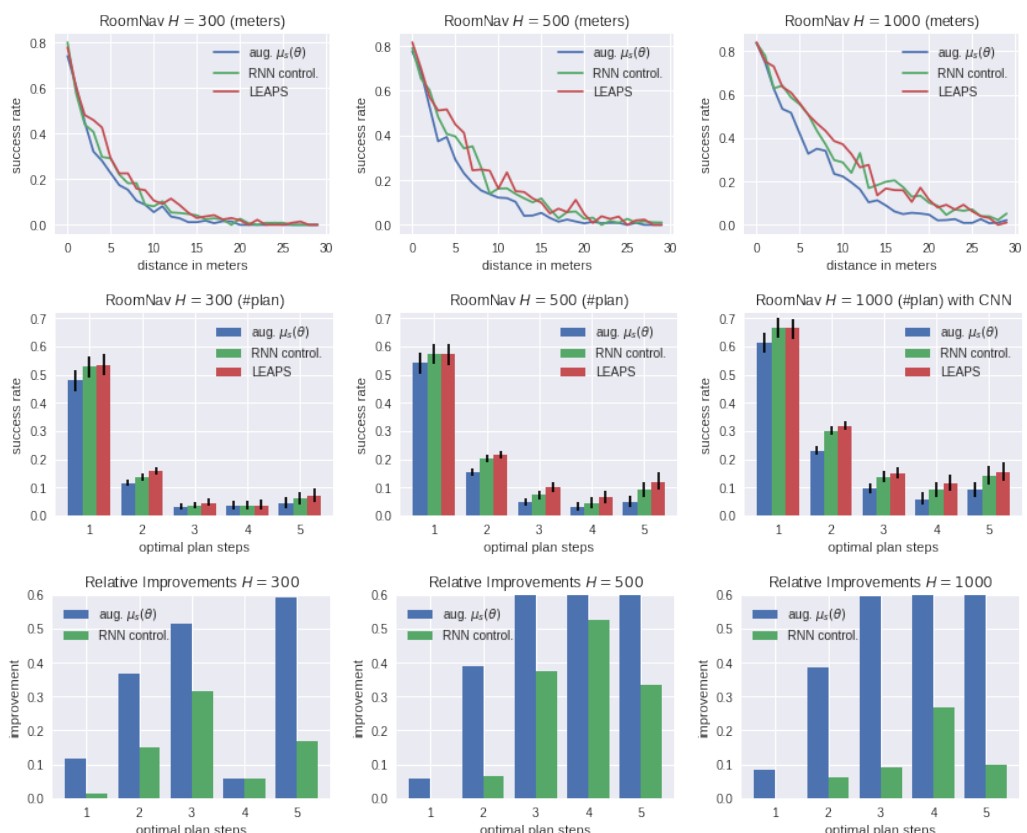

Figure 4: Comparison in success rate with semantic-aware policies (Sec. 6.3). We evaluate performance of the semantic augmented model-free agent ("aug. $\mu_s(\theta)$", blue), the HRL agent with the same sub-policies as LEAPS but with an LSTM controller ("RNN control.", green) and our LEAPS agent (red), with increasing horizon $H$ from left to right (**left**: $H = 300$; **middle**: $H = 500$; **right**: $H = 1000$). **Top row**: success rate (y-axis) w.r.t. the distance in meters from birthplace to target (x-axis); **middle row**: success rate with confidence interval (y-axis) w.r.t. the shortest planning distance in the ground truth semantic model (x-axis); **bottom row**: relative improvements of LEAPS over the baselines (y-axis) w.r.t. the optimal plan distance (x-axis). Our LEAPS agent outperforms both of the baselines for targets requring planning computations (i.e., plan-steps > 1). For faraway targets with plan-steps > 2 in longer horizons ($H \geq 500$), LEAPS improves over augmented policy by **80%** and over RNN controller by **10%** in success rate. Note that even though the LSTM controller has two orders of magnitudes more parameters than our semantic model $\mathbf{M}$, our LEAPS agent still performs better.

baselines strictly increases. This again indicates the effectiveness of our semantic model and shows that it indeed helps solve harder tasks, i.e., finding those faraway targets requiring more planning computations.

We also notice that in shorter horizons ($H \leq 500$), for plan-steps equal to 4, LEAPS agents have the highest success rate but relatively lower SPL. This is because that our LEAPS agent updates the semantic model every fixed $N = 30$ steps. This relatively low update frequency may potentially increase the episode length to reach a goal that requires more planning computations. However, when the horizon is long, i.e., allowing enough planning computations, our LEAPS agents significantly outperform all the baselines in SPL metric. It will be helpful if the LEAPS agent can learn to update the semantic model instead of updating it in a fixed frequency. We leave this to future works.

# 7    CONCLUSION AND FUTURE WORK

In this work, we proposed LEAPS to improve generalization of RL agents in unseen environments with diverse room layouts and object arrangements, while the underlying semantic information is

| opt plan-steps | 1 | 2 | 3 | 4 | 5 | overall |
|---|---|---|---|---|---|---|
| Horizon $H = 300$ | | | | | | |
| random | 20.5 / 15.9 | 6.9 / 16.7 | 3.8 / 10.7 | 1.6 / 4.2 | 3.0 / 8.8 | 7.2 / 13.6 |
| pure $\mu(\theta)$ | 49.4 / 47.6 | 11.8 / 27.6 | 2.0 / 4.8 | 2.6 / **10.8** | 4.2 / 13.2 | 13.1 / 22.9 |
| aug.$\mu_S(\theta)$ | 47.8 / 45.3 | 11.4 / 23.1 | 3.0 / 7.8 | 3.4 / 8.1 | 4.4 / 11.2 | 13.0 / 20.5 |
| RNN control. | 52.7 / 45.2 | 13.6 / 23.6 | 3.4 / 9.6 | 3.4 / 10.2 | 6.0 / 17.6 | 14.9 / 21.9 |
| LEAPS | **53.4 / 58.4** | **15.6 / 31.5** | **4.5 / 12.5** | **3.6** / 6.6 | **7.0 / 18.0** | **16.4 / 27.9** |
| Horizon $H = 500$ | | | | | | |
| random | 21.9 / 16.9 | 9.3 / 18.3 | 5.2 / 12.1 | 3.6 / 6.1 | 4.2 / 9.9 | 9.1 / 15.1 |
| pure $\mu(\theta)$ | 54.0 / 57.5 | 15.9 / 25.6 | 3.8 / 7.7 | 2.8 / 6.4 | 4.8 / 8.6 | 16.2 / 22.9 |
| aug.$\mu_S(\theta)$ | 54.1 / 51.8 | 15.5 / 26.5 | 4.6 / 8.2 | 3.0 / **11.8** | 4.6 / 12.5 | 16.1 / 23.5 |
| RNN control. | **57.4** / 43.8 | 20.2 / 28.0 | 7.2 / 14.6 | 4.2 / 8.0 | 9.0 / 16.0 | 19.9 / 24.6 |
| LEAPS | 57.2 / **61.9** | **21.5 / 34.4** | **10.0 / 14.8** | **6.4** / 11.6 | **12.0 / 23.5** | **21.6 / 31.1** |
| Horizon $H = 1000$ | | | | | | |
| random | 24.3 / 17.6 | 13.5 / 20.3 | 9.1 / 14.3 | 8.0 / 9.3 | 7.0 / 11.5 | 13.0 / 17.0 |
| pure $\mu(\theta)$ | 60.8 / **58.4** | 23.3 / 29.5 | 7.6 / 8.8 | 8.2 / 12.9 | 11.0 / 17.2 | 22.5 / 26.5 |
| aug.$\mu_S(\theta)$ | 61.3 / 50.1 | 23.0 / 26.2 | 9.4 / 12.0 | 5.8 / 9.6 | 9.0 / 13.6 | 22.4 / 23.8 |
| RNN control. | **66.7** / 49.0 | 30.1 / 31.5 | 13.8 / 15.4 | 9.0 / 10.0 | 14.0 / 20.8 | 28.2 / 27.7 |
| LEAPS | 66.4 / **58.4** | **31.9 / 40.5** | **15.0 / 18.3** | **11.4 / 17.0** | **15.4 / 27.1** | **29.7 / 35.2** |

Figure 5: Metrics of ***Success Rate(%) / SPL(‰)*** evaluating the performances of LEAPS and baseline agents. Our LEAPS agents have the highest success rates for all the cases requiring planning computations, i.e., plan-steps larger than 1. For SPL metric, LEAPS agents have the highest overall SPL value over all baseline methods (rightmost column). More importantly, as the horizon increases, LEAPS agents outperforms best baselines more. LEAPS requires a relatively longer horizon for the best practical performances since the semantic model is updated every fixed $N = 30$ steps, which may potentially increase the episode length for short horizons. More discussions are in Sec. 6.4.

shared with the environments in which the agent is trained on. We adopt a graphical model over semantic signals, which are low-dimensional binary vectors. During evaluation, starting from a prior obtained from the training set, the agent plans on model, explores the unknown environment, and keeps updating the semantic model after new information arrives. For exploration, sub-policies that focus on multiple targets are pre-trained to execute primitive actions from visual input. The semantic model in LEAPS is lightweight, interpretable and can be updated dynamically with little explorations. As illustrated in the House3D environment, LEAPS works well for environments with semantic consistencies – typical of realistic domains. On random environments, e.g., random mazes, LEAPS degenerates to exhaustive search.

Our approach is general and can be applied to other tasks, such as robotics manipulations where semantic signals can be status of robot arms and object locations, or video games where we can plan on semantic signals such as the game status or current resources. In future work we will investigate models for more complex semantic structures.

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

## A  COMPLETE NOTATIONS AND DEFINITIONS

**Environment:** We consider a *contextual Markov Decision Process* (Hallak et al., 2015) $E(c)$ defined by $E(c) = (\mathcal{S}, \mathcal{A}, P(s'|s, a; c), r(s, a; c))$, where $\mathcal{S}$ is the state space and $\mathcal{A}$ is the action space. $c$ represents the objects, layouts and any other *semantic* information describing the environment, and is sampled from $\mathcal{C}$, the distribution of possible semantic scenarios. $r(s, a; c)$ denotes the reward function while $P(s'|s, a; c)$ describes transition probability conditioned on $c$. For example, $c$ can be intuitively understood as encoding the complete map for navigation, or the complete object and obstacle layouts in robotics manipulations, not known to the agent in advance, and we refer to them as the *context*.

**Semantic Signal:** At each time step, the agent's observation is a tuple $(s_o, s_s)$, which consists of: (a) a high-dimensional observation $s_o$, e.g. the first person view image, and (b) a low-dimensional *semantic signal* $s_s$, which encodes semantic information. Such low-dimensional discrete signals are commonly used in AI, e.g. in robotic manipulation tasks $s_s$ indicates whether the robot is holding an object; for games it is the game status of a player; in visual navigation it indicates whether the agent reached a landmark; while in the AI planning literature, $s_s$ is typically a list of *predicates* that describe binary properties of objects. We assume $s_s$ is provided by an oracle function, which can either be directly provided by the environment or extracted by some semantic extractor.

**Generalization:** Let $\mu(a|\{s^{(t)}\}_t; \theta)$ denote the agent's policy parametrized by $\theta$ conditioned on the previous states $\{s^{(t)}\}_t$ and $R(\mu(\theta); c)$ denote the accumulative reward of $\mu(\theta)$ in $E(c)$. The objective is to find the best policy that maximizes the *expected* accumulative reward $\mathbb{E}_{c \sim \mathcal{C}}\left[R(\mu(\theta); c, T_i)\right]$. In practice, we sample a disjoint partition of a training set $\mathcal{E}_{\text{train}} = \{E(c_i)\}_i$ and a testing set $\mathcal{E}_{\text{test}} = \{E(c_j)\}_j$, where $\{c_i\}$ and $\{c_j\}$ are samples from $\mathcal{C}$. We train $\mu(\theta)$ with a *shaped* reward $r_{\text{train}}$ only on $\mathcal{E}_{\text{train}}$, and measure the *empirical* generalization performance of the learned policy on $\mathcal{E}_{\text{test}}$ with the original unshaped reward (e.g., binary reward of success or not).

## B  ENVIRONMENT DETAILS

In RoomNav the 8 targets are: kitchen, living room, dining room, bedroom, bathroom, office, garage and outdoor. We inherit the success measure of "see" from (Wu et al., 2018): the agent needs to see some corresponding object for at least 450 pixels in the input frame and stay in the target area for at least 3 time steps.

For the binary signal $s_s$, we obtain from the bounding box information for each room provided from SUNCG dataset (Song et al., 2017), which is very noisy.

Originally the House3D environment supports 13 discrete actions. Here we reduce it to 9 actions: large forward, forward, left-forward, right-forward, large left rotate, large right rotate, left rotate, right rotate and stay still.

## C  EVALUATION DETAILS

We following the evaluation setup from (Wu et al., 2018) and measure the success rate on $\mathcal{E}_{\text{test}}$ over 5750 test episodes, which consists of 5000 random generated configurations and 750 specialized for faraway targets to increase the confidence of measured success rate. These 750 episodes are generated such that for each plan-distance, there are at least 500 evaluation episodes. Each test episode has a fixed configuration for a fair comparison between different approaches, i.e., the agent will always start from the same location with the same target in that episode. Note that we always ensure that (1) the target is connected to the birthplace of the agent, and (2) the the birthplace of the agent is never within the target room.

## D  EVALUATION RESULTS USING GROUND TRUTH SEMANTIC SIGNALS

In the experiment sections, we use a CNN detector to extract the semantic signals at test time. Here we also evaluate the performance of all the approaches when using the ground truth signal from the oracle provided by the House3D environment. The results are in Figure 6, where we also include the LEAPS agent using CNN detector as a references. Generally, using both the ground truth signal and using the CNN detector yield comparable overall performances in both metrics of success rate and

SPL. They all consistently outperform all the baseline methods, which indicates that our Bayesian model is indeed robust over semantic signals. One interesting observation is that there are many cases, using CNN detector produces better results than using the ground truth signals. We hypothesis that this is because the semantic labels in House3D is noisy and therefore a well-trained CNN detector will not be influenced by the noisy labels at test time.

| plan-dist | 1 | 2 | 3 | 4 | 5 | overall |
|---|---|---|---|---|---|---|
| Horizon $H = 300$ | | | | | | |
| random | 20.5 / 15.9 | 6.9 / 16.7 | 3.8 / 10.7 | 1.6 / 4.2 | 3.0 / 8.8 | 7.2 / 13.6 |
| pure $\mu(\theta)$ | 49.4 / 47.6 | 11.8 / 27.6 | 2.0 / 4.8 | 2.6 / 10.8 | 4.2 / 13.2 | 13.1 / 22.9 |
| aug.$\mu_S(\theta)$ (true) | 51.9 / 66.4 | 11.1 / 24.2 | 3.3 / 7.8 | 2.4 / 6.0 | 3.0 / 8.7 | 13.2 / 23.3 |
| RNN contrl. (true) | 52.9 / 44.7 | 13.9 / 26.2 | 4.7 / 10.4 | 2.0 / **6.6** | 5.4 / 17.1 | 15.2 / 22.9 |
| LEAPS (true) | **54.1 / 67.4** | **15.9 / 34.1** | **6.1 / 15.1** | 2.8 / 5.4 | 6.2 / **22.1** | **16.7 / 31.0** |
| LEAPS (CNN) | 53.4 / 58.4 | 15.6 / 31.5 | 4.5 / 12.5 | **3.6 / 6.6** | **7.0** / 18.0 | 16.4 / 27.9 |
| Horizon $H = 500$ | | | | | | |
| random | 21.9 / 16.9 | 9.3 / 18.3 | 5.2 / 12.1 | 3.6 / 6.1 | 4.2 / 9.9 | 9.1 / 15.1 |
| pure $\mu(\theta)$ | 54.0 / 57.5 | 15.9 / 25.6 | 3.8 / 7.7 | 2.8 / 6.4 | 4.8 / 8.6 | 16.2 / 22.9 |
| aug.$\mu_S(\theta)$ (true) | 55.1 / 58.3 | 15.3 / 23.0 | 4.9 / 8.7 | 2.2 / 5.9 | 6.2 / 15.5 | 16.3 / 22.4 |
| RNN contrl. (true) | **57.2** / 44.9 | 19.5 / 27.4 | 6.2 / 10.3 | 4.2 / 8.6 | 8.4 / 12.3 | 19.3 / 23.3 |
| LEAPS (true) | 57.0 / 59.8 | 21.0 / 33.5 | 9.5 / **17.3** | **6.6** / 10.8 | 10.2 / 22.3 | 21.1 / 30.6 |
| LEAPS (CNN) | **57.2 / 61.9** | **21.5 / 34.4** | **10.0** / 14.8 | 6.4 / **11.6** | **12.0 / 23.5** | **21.6 / 31.1** |
| Horizon $H = 1000$ | | | | | | |
| random | 24.3 / 17.6 | 13.5 / 20.3 | 9.1 / 14.3 | 8.0 / 9.3 | 7.0 / 11.5 | 13.0 / 17.0 |
| pure $\mu(\theta)$ | 60.8 / 58.4 | 23.3 / 29.5 | 7.6 / 8.8 | 8.2 / 12.9 | 11.0 / 17.2 | 22.5 / 26.5 |
| aug.$\mu_S(\theta)$ (true) | 62.4 / 61.3 | 22.9 / 30.7 | 8.9 / 14.3 | 7.2 / 12.8 | 9.0 / 11.4 | 22.5 / 28.1 |
| RNN contrl. (true) | 65.4 / 50.7 | 29.9 / 33.8 | 14.6 / 16.6 | 9.2 / 12.8 | 13.6 / 21.7 | 28.1 / 29.6 |
| LEAPS (true) | **66.5 / 62.3** | **33.8 / 42.0** | **17.8 / 19.6** | 11.0 / 10.2 | **18.4 / 29.8** | **31.4 / 36.3** |
| LEAPS (CNN) | 66.4 / 58.4 | 31.9 / 40.5 | 15.0 / 18.3 | **11.4 / 17.0** | 15.4 / 27.1 | 29.7 / 35.2 |

Figure 6: Metrics of ***Success Rate(%) / SPL(‰)*** evaluating the performances of LEAPS and baselines agents using the ground truth oracle semantic signals provided by the environments. We also include the performance of LEAPS agent using CNN detector as a reference. Note that even using an CNN detector, LEAPS agents outperforms all baselines in both metrics of success rate and SPL. Notably, the performance of LEAPS-CNN agents is comparable to LEAPS-true agents and sometimes even better. This indicates that our semantic model can indeed tolerate practical errors in CNN detectors. More discussions are in Sec. D.

## E    EVALUATION DETAILS ON EPISODE LENGTH

We illustrate the ground truth shortest distance information as well as the average episode length of success episodes for all the approaches. The results are shown in Figure 7. Note that the average ground truth shortest path is around 46.86 steps. Considering the fact agent has 9 actions per step as well as the strong partial observability, this indicates that our benchmark semantic navigation task is indeed challenging.

## F    VISUALIZATION DETAILS

For confidence interval of the measured success rate, we computed it by fitting a binomial distribution.

For optimal plan steps, we firstly extract all the room locations, and then construct a graph where a vertex is a room while an edge between two vertices is the shortest distance between these two rooms. After obtaining the graph and a birthplace of the agent, we compute shortest path from the birthplace to the target on this graph to derive the optimal plan steps.

| Average Ground Truth Shortest Path Length | | | | | | |
|---|---|---|---|---|---|---|
| plan-dist | 1 | 2 | 3 | 4 | 5 | overall |
| Oracle | 12.27 | 42.53 | 61.09 | 72.47 | 63.74 | 46.86 |
| Average Successful Episode Length | | | | | | |
| plan-dist | 1 | 2 | 3 | 4 | 5 | overall |
| Horizon $H = 300$ | | | | | | |
| random | 34.0 | 112.7 | 143.8 | 148.0 | 149.7 | 89.8 |
| pure $\mu(\theta)$ | 55.2 | 107.0 | 127.9 | 140.8 | 139.4 | 84.7 |
| aug.$\mu_S(\theta)$ | 49.7 | 112.5 | 159.9 | 179.1 | 176.8 | 89.2 |
| RNN control. | 61.6 | 122.9 | 127.1 | 131.0 | 124.5 | 96.2 |
| LEAPS | 58.3 | 120.6 | 160.1 | 178.6 | 154.7 | 99.7 |
| Horizon $H = 500$ | | | | | | |
| random | 56.1 | 186.0 | 206.5 | 286.7 | 222.0 | 154.1 |
| pure $\mu(\theta)$ | 74.1 | 184.6 | 205.9 | 190.6 | 215.6 | 140.4 |
| aug.$\mu_S(\theta)$ | 86.7 | 178.1 | 240.6 | 185.5 | 267.3 | 145.6 |
| RNN control. | 94.0 | 206.4 | 237.7 | 252.0 | 256.3 | 171.0 |
| LEAPS | 83.1 | 203.9 | 267.6 | 258.3 | 248.6 | 173.1 |
| Horizon $H = 1000$ | | | | | | |
| random | 121.7 | 354.7 | 426.6 | 532.8 | 409.5 | 322.1 |
| pure $\mu(\theta)$ | 145.8 | 378.1 | 504.5 | 491.2 | 458.2 | 315.1 |
| aug.$\mu_S(\theta)$ | 163.1 | 360.9 | 471.9 | 460.7 | 432.5 | 307.1 |
| RNN control. | 170.0 | 407.4 | 512.7 | 485.0 | 419.5 | 350.0 |
| LEAPS | 158.0 | 379.2 | 512.0 | 488.1 | 419.4 | 335.9 |

Figure 7: Averaged successful episode length for different approaches. The length of shortest path reflects the strong difficulty of this task.

## G    DETAILS FOR LEARNING NEURAL SUB-POLICIES

**Hyperparameters:**  We utilize the same policy architecture as (Wu et al., 2018). It was mentioned in (Wu et al., 2018) that using segmentation mask + depth signals as input leads to relatively better performances for policy learning. So we inherit this setting here. In the original House3D paper, a gated attention module is used to incorporate the target instruction. Here, since we only have $K = 8$ different sub-policies, we simply train an individual policy for each target and we empirically observe that this leads to better performances. We run A3C with $\gamma = 0.97$, batch size 64, learning rate 0.001 with Adam, weight decay $10^{-5}$, entropy bonus 0.1. We backprop through at most 30 time steps. We also compute the squared $l_2$ norm of logits and added to the loss with a coefficient 0.01. We also normalize the advantage to mean 0 and standard deviation 1.

**Reward shaping:**  We used a shaped reward function similar to (Wu et al., 2018): the reward at each time step is computed by the difference of shortest paths in meters from the agent's location to the goal after taking a action. We also add a time penalty of 0.1 and a collision penalty of 0.3. When the agent reaches the goal, the success reward is 10.

**Curriculum learning:**  We run a curriculum learning by increasing the maximum of distance between agent's birth meters and target by 3 meters every 10000 iterations. We totally run 60000 training iterations and use the final model as our learned policy $\mu(\theta)$.

## H    DETAILS FOR LEARNING THE SEMANTIC MODEL

After evalution on the validation set, we choose to run random exploration for 300 steps to collect a sample of $z$. For a particular environment, we collect totally 50 samples for each $z_{i,j}$.

For all $i \neq j$, we set $\psi_{i,j,0}^{\text{obs}} = 0.001$ and $\psi_{i,j,1}^{\text{obs}} = 0.15$.

## I    ADDITIONAL DETAILS FOR TRAINING SEMANTIC-AWARE POLICIES

For the LSTM controller, we ran A2C with batch size 32, learning rate 0.001 with adam, weight decay 0.00001, gamma 0.99, entropy bonus 0.01 and advantage normalization. The reward function is designed as follows: for every subtask it propose, it gets a time penalty of 0.1; when the agent reach the target, it gets a success bonus of 2.

The input of the LSTM controller consists of (1) $s_s{}^{(t)}$ ($K$ bits), (2) $B$ ($K$ bits), (3) last subtask $T_k$, and (4) the final target $T_i$. We convert $T_i$ and $T_k$ to a one-hot vector and combine the other two features to feed into the LSTM. Hence the input dimension of LSTM controller is $4K$, namely 32 in RoomNav.

For the semantic augmented LSTM policy, $\mu_s(\theta_s)$, we firstly use the CNN extract visual features from $s_o$ and combine the input semantic features and the visual features as the combined input to the LSTM in the policy.

## J    ADDITIONAL DETAILS FOR TRAINING THE CNN SEMANTIC EXTRACTOR

We noticed that in order to have a room type classifier, only using the single first person view image is not enough. For example, the agent may face towards a wall, which is not informative, but is indeed inside the bedroom (and the bed is just behind).

So we take the panoramic view as input, which consists of 4 images, $s_o^1, \ldots, s_o^4$ with different first person view angles. The only exception is that for target "outdoor", we notice that instead of using a panoramic view, simply keeping the recent 4 frames in the trajectory leads to the best prediction accuracy. We use an CNN feature extractor to extract features $f(s_o^i)$ by applying CNN layers with kernel size 3, strides $[1, 1, 1, 2, 1, 2, 1, 2, 1, 2]$ and channels $[4, 8, 16, 16, 32, 32, 64, 64, 128, 256]$. We also use relu activation and batch norm. Then we compute the attention weights over these 4 visual features by $l_i = f(s_o^i) W_1^T W_2 \left[ f(s_o^1), \ldots, f(s_o^4) \right]$ and $a_i = \text{softmax}(l_i)$. Then we compute the weighted average of these four frames $g = \sum_i a_i f(s_o^i)$ and feed it to a single layer perceptron with 32 hidden units. For each semantic signal, we generate 15k positive and 15k negative training data from $\mathcal{E}_{\text{train}}$ and use Adam optimizer with learning rate $5e\text{-}4$, weight decay $1e\text{-}5$, batch size 256 and gradient clip of 5. We keep the model that has the best prediction accuracy on $\mathcal{E}_{\text{valid}}$.

For a smooth prediction during testing, we also have a hard threshold and filtering process on the CNN outputs: $s_s(T_i)$ will be 1 only if the output of CNN has confidence over 0.85 for consecutively 3 steps.

