# OpenReview forum: "Learning and Planning with a Semantic Model"
_ICLR.cc/2019/Conference_

### Official Review · AnonReviewer3 · 2018-10-17
**non-effective method (works well only with groundtruth information), convoluted writing, improper evaluation metric**

**Rating:** 5
**Confidence:** 4

**Review:**

The paper proposes a hybrid model-free and model-based RL agent for the task of navigation. Reaching the target is decomposed into a set of sub-goals, and the plan is updated as the agent explores the environment. The method has been tested in the House3D environment for the task of RoomNav, where the goal is to navigate towards a certain room.

The idea of integrating RL agents with semantic knowledge is interesting. However, the paper has several major issues that should be addressed in the rebuttal:

(1) The experiment results in Figure 3 and Figure 4 are based on groundtruth room information. The only experiment that is fully automatic is the one in Figure 5. However, there is no difference between the proposed method and the baselines in that case. So the proposed method is not effective without groundtruth information.

(2) The only evaluation metric that is used is "Success Rate". That metric is not sufficient for evaluation of navigation agents since it does not include episode length information. All of the results should be based on the protocol mentioned in "On Evaluation of Embodied Navigation Agents", arXiv 2018.

(3) There is no termination action according to Appendix B. So the agent does not know if it is at the target or not. It seems the agent will stop if it issues "stay still" three times. That is different from termination action. Also, it is confusing what 450 pixels means for a scene classifier that works on the image.

(4) The paper is written in a convoluted way:
   (a) It is not clear if the semantic model is trained along with the RL model end-to-end or not.
   (b) Regarding multi-target sub-policies, is there a separate policy for each pair of intermediate targets?
   (c) Regarding inference and planning on M, what is \tau exactly? How is the length of the plan determined?
   (d) Why is the model updated only after a fixed number of steps? That increases the episode length.

(5) The number of T_i's is manually set to 8. That causes serious generalization issues. How do we know how many T_i's exist in a new environment?


Minor comments:
- The paper mentions "An example of such environments is House3D which contains 45k real-world 3D scenes". House3D includes only synthetic scenes. They should not be called real-world scenes.
- How is the reward shaping done?

****
Final comments after reading the response and the reviews:

Regarding the fairness of the review, success rate is not sufficient to evaluate navigation agents. A random agent can achieve 100% success if it is given enough time. So it is totally fair to ask for a metric (such as SPL) that is a function of both success rate and episode length.

I am going to increase the rating to 5 since some of my concerns have been addressed. There are still a number of issues:

- The authors did not run the experiments with the termination action. I disagree that this is orthogonal to the focus of the paper. This is not just an additional action. It indicates whether the agent has learned anything or it is just a combination of better obstacle avoidance and luck. The SPL numbers are so low (maximum SPL is 6.19%) so adding the termination action will probably make the method similar to random.

- There is a huge gap between success rate and SPL numbers. For instance, success rate is 66.4, while SPL is 5.84 (note that for some reason the SPL numbers are multiplied by 10 in the table). I doubt that the agent has learned anything meaningful in comparison to the baseline. I understand that the task is hard, but this gap is so huge.

- A separate policy is trained for each sub-target. This doesn't scale. There should be one policy for all targets.

---

> ### Author Response · Authors · 2018-11-10
> **We have added results with SPL (Fig.5), LEAPS is much more effective under SPL metric.**
>
>
> >>> groundtruth information:
> We have replaced all the plots in the updated paper with CNN-LEAPS, namely in the current version, **all the signals are extracted by CNN**. For a clear comparison, we illustrate the success rate in numbers in *Figure 5*. LEAPS has the highest success rate overall and for all targets requiring planning computations (i.e., plan-steps > 1). For targets with plan-steps = 1, they  require little planning so it is as expected that improvements by LEAPS is tiny.
>
> We also show in Appendix D that CNN-LEAPS has comparable performances to LEAPS using ground truth signals.
>
> >>> evaluation metric
> Thanks for pointing out this metric. We agree that this is a better metric and we have evaluated the performance of all approaches under SPL in Sec 6.4. All the results are shown in Figure 5.
> ** Surprisingly, under SPL, LEAPS has very significant overall improvements over all the baseline methods in all the horizons (rightmost column in Figure 5). ** Particularly, as the horizon becomes longer, namely with more allowed planning steps, the margin between LEAPS and other baselines are increasingly larger. Which again indicates that LEAPS is indeed able to solve faraway targets.
> Although we appreciate the reviewer for pointing out the paper “On Evaluation of Embodied Navigation Agents”, we think it is unfair to say “All of the results *should* be based on the protocol mentioned in” this paper. It is a very recent paper (on arxiv in Jul) while our project started much earlier. Also, we think as a research work, it is still debatable to say which metric is better or worse (e.g., the paper itself uses the term "recommend"). We are not required to follow this unpublished work (though it is indeed insightful). Nevertheless, we have already added results under SPL metrics and hope that Reviewer 3 is happy with these results.
>
>
> >>> termination action
> The design of action space is an orthogonal issue to our focus. Here we propose a HRL framework improve generalization ability of DRL agents on environments with semantic regularities. For this purpose, we choose the RoomNav task on House3D environment. Even without a termination action in the action space, the task itself is already very challenging due to two factors: (1) long horizon and large action space (see Appendix F for details), i.e., avg shortest path is 46.86 steps away and the agent has 9 actions; (2) strong partial observability, i.e., many existing navigation works are within the same room while our task needs to navigate through rooms --- visual signals of other rooms are blocked by walls and doors.
> We agree that ultimately for building a navigation agent, we should eventually include termination action in the action space, as suggested by the paper “On Evaluation of Embodied Navigation Agents”. However, we again emphasize that this is orthogonal to our focus (we are proposing an learning framework), and since this paper is very recent and unpublished, we think it is unfair to complain us on the design of action space according to it.
>
> >>> writing
> Thanks for the comments and we updated the paper accordingly.
> (a) We have added Sec 4.4 in the main paper to further clarify that LEAPS is trained in a two-step fashion. First sub-policies and then semantic models.
> (b) it is a conditional LSTM policy conditioning on memory, current visual input and the semantic target.
> (c) We updated the texts in Sec 4.2. tau denotes a sequence of concepts that leads to the goal concept with the highest probability. The length varies according to the planning result on the semantic graph.
> (d) The semantic model should be updated periodically. Updating it after a fixed amount of steps is the easiest way. We found this simple approach generally works well. Yes, it may indeed increase the episode length. This is why in some cases under short horizon LEAPS may not have the highest SPL. We carefully discussed this in Sec 6.4.
>
> >>> size of concepts
> We can never expect a human who never know the concept of apple before to solve a semantic task “find an apple”. It is analog to natural language where each word in the agent’s dictionary can be considered as a concept here. The T_i’s can be understood as the word tokens the agent understands. In environments with man-made semantic structures, these concepts (T_i’s) generalize. If at test time we cannot categorize some scenes into our known concepts, it is completely fine. Practically, we do have a concept called “other” in our implementation (see footnote 1) representing all those scenes not belonging to any of predefined concepts in our dictionary.
>
> Minor comments:
> -- we have updated the introduction section accordingly
> -- we added description to reward shaping in Appendix G.

---

> > ### Comment · AnonReviewer3 · 2018-11-19
> > **SPL results seem incorrect**
> >
> > Thanks for addressing my comments. However, the SPL results do not seem correct. Success rate should be the upperbound for SPL so SPL cannot be higher than success rate, but in Figure 5 there are several cases that SPL is higher than success rate. Please fix the results so we can better evaluate the paper. We just need to see the episode length. Any metric that considers both episode length and success rate together would work. SPL is one option.

---

> > > ### Author Response · Authors · 2018-11-19
> > > **SPL results are correct.**
> > >
> > > Thanks for the concern. Our SPL results are correct. Note that in Table. 5 (or Fig. 5) we report success rate as percentage (in the unit of 0.01), and report SPL as per-mille (in the unit of 0.001). For example, 53.4 / 58.4 in the table means that the success rate is 53.4% while SPL is 0.0584. So SPL is indeed upper-bounded by success rate, as suggested by the reviewer (and by the definition of SPL).
> > >
> > > Overall, SPL is low for all the methods, due to the difficulty of our navigation task (it is multi-room, and unknown environment to the agent).
> > >
> > > For episode length, please check Table. 7 (or Fig. 7).

---

### Official Review · AnonReviewer2 · 2018-11-03
**This paper proposes a hybrid model-based and model-free approach called LEAPS, consisting of a multi-target sub-policy that acts on visual inputs, and a Bayesian model over semantic structures.**

**Rating:** 7
**Confidence:** 3

**Review:**

The contributions of this paper are in the area of semantic modelling, where the authors propose an approach called LEAPS consisting of a multi-target sub-policy that acts on visual inputs, and a Bayesian model over semantic structures. The fundamental premise of the proposed approach is that when placed in an unseen environment the agent plans with the semantic model based on new observations. Particularly, the authors propose to learn a Bayesian model over the semantic level and infer the posterior structure via the Bayes rule. The proposed approach is validated with experiments in visual navigation tasks using a 3D environment that contains diverse human-designed indoor scenes with real world objects. Finally, the authors show the key role of using semantic context compared to the baselines that do not consider semantic context.

The parer is interesting, well structured and and clearly written. Also, the addressed topic of incorporating semantic model in the context of learning and planning is very interesting.

The related work is extensively presented with pertinent and up-to-date literature. Furthermore, the background section presents well the DRL notations.

In section 5, how the values for e.g between dinning room and garage 0.05, dinning room and kitchen 0.7 are learned, and how generalisable is this approach to other applications - because the way those priors are determined do not seem very explicit?

Furthermore in the experiments it does not seem explicit how the semantic model is updated in light of new information, I think this deserves further explanation or to be clearly pinpointed?

Also, what are the key requirements that make  the semantic model interpretable. Because, the way the validation is conducted in this paper, it seems that ithe nterpretability is quite specific to House3D - is it generalisable to other applications and under which conditions?

Otherwise, I believe that the questions asked in the experiments section are well answered with the experimental results

---

> ### Author Response · Authors · 2018-11-10
> **Thanks for the valuable comments**
>
> The prior distribution is learned from our training environments by random exploration. This semantic prior can be generally applied to tasks that require reasoning over the relations between rooms.
> Our approach can be applied to other tasks and applications involving semantic properties and relations as well. The Bayesian framework as well as the learning algorithm is general. When a new domain, we just define the Bayesian model over concepts in the new domain and the overall framework still works.
> As stated in Sec 6.2, we update our semantic model every 30 time steps. A show case is in Figure 2 where we show the updated graph with posterior probabilities on each edge.

---

### Official Review · AnonReviewer1 · 2018-11-05
**Learning and Planning with a Semantic Model**

**Rating:** 4
**Confidence:** 4

**Review:**

This work proposes a hybrid model for robot visual navigation in synthetic indoor environments, specifically a combination of a  high-level planning scheme (model-based) with a low-level behavior based approach (model-free) . The main contribution is on the high-level based planning that is based on semantic cues from the environment, specifically the construction of a semantic prior about rooms connectivity. By using this prior the system is able to generalize to new environments simplifying an initial robot exploration phase.

The semantic prior is implemented by the construction of a graph representation that encodes room connectivity. Links between rooms (nodes) are given by Bernoulli variables which are inferred by previous experiences and an exploration phase in the current environment.

Results are one of the weaker parts of the paper, success rates are very low, even for short planning horizons (figures 3,4,5).  Furthermore, it is not clear the real relevance of the semantic prior because relative performance with respect to baselines is not significant. In general, while a room connectivity prior can be of help, I believe is not so critical for indoor robot navigation. There are prior works on Robotics that has shown more impact using structural priors, such as, presence of corridors, doors, etc, or "object-room" spatial relations. The low success rate is even more critical if one considers that the validation is based on synthetic environments.

In general the paper is easy to follow, although, there are some details missing, specially in terms of model description. My main concern is that the paper is limited in technical novelty and it suffers from a lack of practical significance.

---

> ### Author Response · Authors · 2018-11-10
> **Please check Fig.5 (new), experiments show that LEAPS is effective**
>
> >>>> For low success rate:
> It is because this semantic navigation task itself is extremely challenging for RL. Therefore we proposed an HRL framework, i.e., a semantic model, to improve performance of general RL methods.
> We emphasize that the low success rate is not training performance. Instead, it is generalization performance on unseen environments. Current state-of-the-art results on semantic navigation tasks still suffer from this low success rate (e.g., fig7 in https://arxiv.org/pdf/1609.05143.pdf, row 3 in table 1 in https://arxiv.org/pdf/1810.06543.pdf).
> Another misunderstood point is that the “planning horizons”  here are steps in the “semantic model”, i.e., the number of rooms the agent needs to go through to reach the goal. It is NOT the actual episode length for the agent. We show the stats of averaged length of ground truth shortest path in Appendix F, where the avg length is 46.86.  Note the agent has 9 available actions and the environment has strong partial observability (visual signal of different rooms are typically blocked by walls). The task is indeed challenging. Moreover, our goal is to improve the performance of model-free RL approaches. So we believe the relative improvement instead of the success rate itself should be focused more on.
>
> >>>> Improvement is not significant:
> To better illustrate that our method is effective, we illustrate all the success rate in numbers in Figure.5. In all the horizons, LEAPS agents have the best success rates overall and for all targets requiring planning computations (i.e., plan-steps > 1). Particularly, the relative improvements are higher for targets requiring more planning (e.g., plan-steps = 3)
> Thanks to the suggestion by Reviewer3, we utilize a better metric, Success weighted by Path Length (SPL), which considers episode length in the evaluation metric. In SPL, success episodes with faraway targets will be assigned more credits. We introduce SPL in Sec 6.4 and show the results in Figure 5. In SPL, our approach overall outperforms all the baselines with significant margins. More importantly, as more planning computations (longer horizons), the margin becomes increasingly larger. This again indicates that LEAPS is indeed effective for those faraway targets.
>
> >>>>  For synthetic dataset:
> Sorry for the confusion. We have updated the introduction section accordingly. We here are refering to environments that resemble the real-world semantic properties. The houses in House3D are designed by humans and share the same semantic regularities as the real world.

---

### Author Response · Authors · 2018-11-10
**We have revised our paper with more experiments and clarifications**

We have updated the paper to accommodate reviewers’ concerns, including many changes in experiment section and additional evaluations with a new **metric SPL (Sec 6.4, Figure 5)**.

Here are details of what we changed.
(1) Improved introduction section (Sec.1) and algorithm section (Sec. 4) incorporating reviewer’s comments
(2) Reviewers are complaining about we use ground truth semantic signals in the experiments. In the current version, all the results of LEAPS in experiment section (Sec.6) are **using CNN ** to extract semantic signals. The case of ground truth signals are deferred to appendix D. We emphasize that our LEAPS agent is robust to semantic signals and has comparable performances with different semantic signals. All our previous claims remain true.
(3) Reviewer3 suggests us to use a more informative metric, **SPL**, which considers episode length information, in addition to success rate. We evaluate LEAPS under SPL in Sec 6.4 (Fig.5). Surprisingly, under SPL metric, LEAPS outperforms all the baselines much **more significantly**. Please check Sec 6.4 for details.
(4) To better illustrate the effectiveness of LEAPS, we show the success rate and SPL in numbers in Figure.5. LEAPS agents have overall better performances under both metrics.

---

### Meta-Review · Area_Chair1 · 2018-12-16
**novel approach to combine model-based and model-free RL - needs more in-depth analysis of results**

**Confidence:** 5
**Recommendation:** Reject

**Metareview:**

The paper presents LEAPS, a hybrid model-based and model-free algorithm that uses a Bayesian approach to reason/plan over semantic features, while low level behavior is learned in a model-free manner. The approach is designed for human-made environments with semantic similarity, such as indoor navigation, and is empirically validated in a virtual indoor navigation task, House3D. Reviewers and AC note the interesting approach to this challenging problem. The presented approach can provide an elegant way to incorporate  domain knowledge into RL approaches.

The reviewers and AC note several potential weaknesses. The reviewers are concerned about the very low success rate, and critiqued the use of success rate as a key metric itself, given that random search with a sufficiently high cut-off could solve the task. The authors added additional results in a metric that incorporates path length, and provided clarifying details. However, key concerns remained given the low success rates. The AC notes that e.g., results in the top and middle row of figure 4 show very similar results for LEAPS and the reported baselines. Further, "figure 5" shows no confidence / error bars, and it is not possible to assess whether any differences are statistically significant. Overall, the questions of whether something substantial has been learned, should be addressed with a detailed error analysis of the proposed approach and the baselines, to provide insight into whether and how the approaches solve the task. At the moment, the paper presents a potentially valuable approach, but does not provide convincing evidence and conceptual insights into the approach's effectiveness.